# A Probabilistic Approach to Optimizing MRI control parameters using Gumbel-Softmax

## Abstract

Optimizing control parameters is crucial to estimate reliable tissue characteristics in quantitative MRI. Basically, multiple hardware parameters are simultaneously controlled to generate a signal from MRI system. Repetitive acquisitions with different control parameter combinations create distinct signal modulations and then tissue characteristics are deduced from prior knowledge of physics-based relationship among modulated signals, control parameters, and tissue characteristics. The choice of control parameters, which determines the attribute of signal modulation, directly impacts the inverse problem in tissue characteristic estimation. Thus, the multidimensional control parameter optimization remains an open research topic in MRI field for accurate analysis of tissue characteristics. Typically, optimal parameters are determined by iteratively updating sets of control parameters to maximize the estimation accuracy of the tissue characteristics. However, the conventional optimization process is restricted to explore only the vicinity of control parameters at the current iteration. Therefore, it could highly depend on initialization and current parameters, which might lead to inefficient search especially when noise is present in the system. In this work, to mitigate this limitation, we propose a novel Gumbel-Softmax-based optimization scheme that enables a probabilistic search across an expanding set of all candidates for each control parameter using categorical reparameterization. As a case study, the proposed method is employed to find optimal control parameters for quantitative MRI. We demonstrate that our Gumbel-Softmax-based optimization simultaneously explores the entire range of control parameters from early iterations and outperforms the conventional optimization approach on accuracy of MR tissue characteristic estimation and repeatability of optimization, especially under noisy environments.

## 1 Introduction

Optimization of control parameters is vital to find accurate tissue characteristics in quantitative MRI. Basically, multiple control parameters are simultaneously adjusted to produce a signal from MRI system. Repetitive acquisitions using various combinations of control parameters yield signal modulations, and tissue characteristics are inferred based on the prior knowledge of the MRI system (Figure 1). Since the observed signal modulations are dependent on control parameters, optimizing control parameters is significant to accurately analyze the tissue characteristics. Various quantitative MRI techniques are designed using their specific physics models, e.g. intravoxel incoherent motion (IVIM), Arterial spin labeling (ASL), Magnetization transfer (MT), etc (Le Bihan (2019); Hilbert et al. (2020); Hernandez-Garcia et al. (2022)). Thus physics model enables simulation of the tissue characteristics through control parameters. Although the physics model could provide an accurate signal, it was challenging to optimize the control parameters due to the high degree of freedom in the physics model and lack of objective function that can capture all aspects of tissue characteristics estimations (Perlman et al. (2023)). Especially, the simulation of physics models in MRI system often includes solving complicated differential equations which does not provide an analytical solution for complex, nonlinear, inverse mapping problem. Therefore, evaluating estimation error of inverse problem for tissue characteristics becomes even more important to ensure an accurate update of control parameters towards optimal solutions.

Recently, the neural network was proposed to solve nonlinear inverse problems of estimating tissue characteristics in quantitative MRI (Hoppe et al. (2017); Yoon et al. (2018); Scannell et al. (2020);

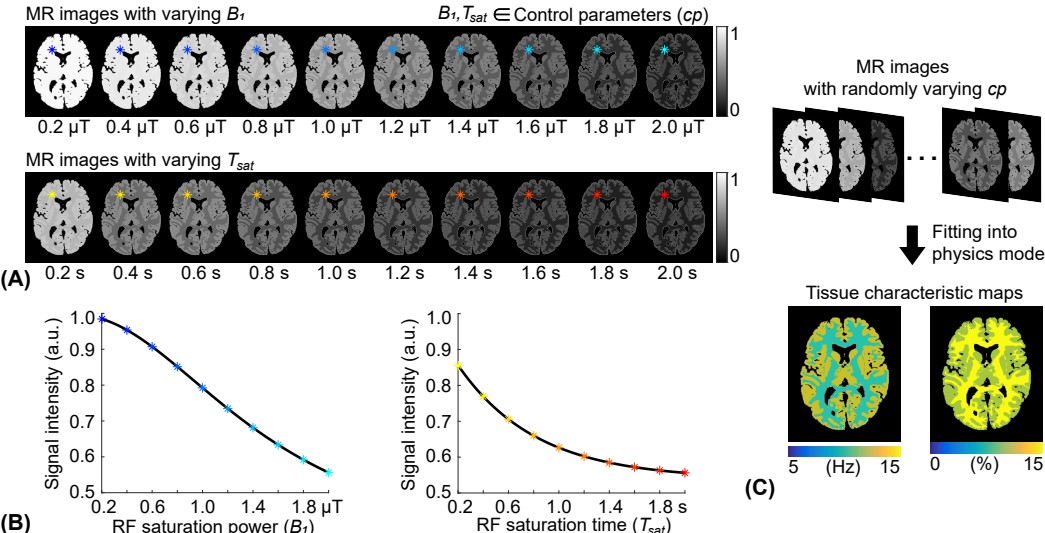

Figure 1: An overview of a quantitative MRI technique. (A) Two series of MR images are obtained with different control parameters ($cp$) of RF saturation power ($B_1$) and saturation time ($T_{sat}$) respectively. (B) Curve fitting is performed to estimate tissue characteristics in a pixel-wise manner. Dots represent acquired signals with varying control parameter and solid curves represent the curve fit for the physics model. (C) A series of MR images acquired with different sets of control parameters, which simultaneously varies $B_1$ and $T_{sat}$, are fitted into physics model to estimate tissue characteristic maps (exchange rate and concentration of semisolid macromolecule for our study).

Jung et al. (2022)). The estimation error could be used as an objective function for optimization and thus the gradient descent technique allowed the control parameters to be updated in a way that minimized the estimation errors of tissue characteristics (Lahiri et al. (2020); Lee et al. (2021); Velasco et al. (2022); Wang et al. (2023)). Learning-based optimization of acquisition schedule (LOAS) has recently been proposed to provide optimal scheduling control parameters for quantitative MRI, directly minimizing estimation errors of tissue characteristics (Kang et al. (2022)). The LOAS algorithm outperformed existing indirect optimization approaches, such as maximizing signal discrimination between tissue types (Cohen & Rosen (2017)) and minimizing the variance of estimates using the Cramer-Rao bound (Zhao et al. (2018)). However, the control parameters have deterministic values at each iteration and updated by exploring only the vicinity of the parameters at the current iteration. As a result, it could heavily rely on initialization and current parameters, potentially leading to inefficient search especially when noise is present in the system. This is also relevant problems with multiple optimal solutions, which are commonly found in real-world scenarios (Huang et al. (2018; 2019); Jian & Hsieh (2022); Xiong et al. (2023)).

To address the aforementioned issues, we propose a novel Gumbel-Softmax-based LOAS (GLOAS) framework that allows a probabilistic search across an expanding set of all candidates for each parameter using categorical reparameterization of MRI control parameters. The Gumbel-Softmax facilitates differentiable categorical reparameterization, enabling a probabilistic representation of control parameters. Thus, the probabilistic representation is updated via gradient descent and eventually control parameters with the highest probability would be selected (Figure 2). To the best of our knowledge, this is the first work that applies categorical reparameterization of control parameters for MRI acqusition optimization.

Our main contributions can be summarized as follows:

- We propose a categorical reparameterization of MRI control parameters with Gumbel-Softmax to allow a probabilistic search across an expanding set of all candidates for each parameter.

- We show that the proposed Gumbel-Softmax-based optimization enables an accurate calculation of gradient for backpropagation in the presence of noise in the system.

- We show that the proposed probabilistic optimization equally explores a wide spectrum of possible control parameters from early iterations, enabling a comprehensive search across the complex multidimensional space of the physics model.

- The experimental results demonstrate the superiority of the proposed method in terms of accuracy of tissue characteristic estimations and repeatability of optimization, especially in the presence of noise for MRI system.

## 2 PHYSICS-MODEL BASED OPTIMIZATION OF CONTROL PARAMETERS

### 2.1 PHYSICS MODEL

For tissue characteristic estimation, the signal modulations obtained from various combinations of control (scan) parameters are fitted into physics model. The physics model is based on the principles of physics that explain the underlying mechanisms of MRI system. Therefore, the physics model can accurately simulate the signal with given tissue characteristics and control parameters. Solving inverse problem of physics model with the known control parameters could provide estimations of tissue characteristic from experimentally observed signal modulations. Basically, the signal ($S$) is defined with two sets of parameters:

$$S = PM(tc, cp) \tag{1}$$

where $tc$ is a set of tissue characteristics and $cp$ is a set of control parameters. Multiple acquisitions are acquired with different combinations of control parameters to generate unique signal modulations which encode the tissue characteristics. Thus, each set of tissue characteristics result in distinct signal modulations, which can be considered as a unique fingerprint for those tissue characteristics. The unique signal modulations with respect to tissue characteristics can be described as follows (Ma et al. (2013); Cohen-Adad et al. (2021); Jordan et al. (2021); Kang et al. (2023)):

$$\mathbf{S}(tc, \mathbf{cp}) = [S(tc, cp_1), ..., S(tc, cp_N)] \tag{2}$$

$$cp_i = [cp_{i,1}, cp_{i,2}, ..., cp_{i,M}] \tag{3}$$

where $N$ represents the number of acquisitions and $M$ is the number of control parameters for single acquisition. Different combinations of $M$ control parameters would result in different signal modulations with the same set of tissue characteristics. Therefore, the choice of control parameters is of importance to discriminate numerous signal modulations which would lead to an accurate estimation of tissue characteristics. For example, linearly-increasing control parameters within the pre-defined range would generate similar signal modulations, regardless of tissue characteristics. This could result in poor discrimination of signal modulations, leading to an inaccurate estimation of tissue characteristics. The control parameters were often chosen to reduce the redundancy between acquisitions (Cohen & Rosen (2017); Kim et al. (2020)).

The physics model is used to understand the complex relation of signal modulations, control parameters, and tissue characteristics. Although the inverse problem of physics-based model is often ill-posed due to its intricacy, many studies have addressed it with neural networks (Cohen et al. (2018); Aggarwal et al. (2018); Jun et al. (2021)):

$$\widehat{tc} = PM^{-1}(\mathbf{S}(tc, \mathbf{cp})) = f_\theta(\mathbf{S}(sp, \mathbf{cp})) \tag{4}$$

where $\widehat{tc}$ is the estimated tissue characteristics by solving the inverse problem, $f$ is a deep neural network, and $\theta$ represents the parameters of the network. However, the inverse problems become even more complicated in real world due to noise. The noise could be originated from various sources such as diverse physiological processes, thermal noise, motions, and many others (Brooks et al. (2013)). Moreover, the control parameter themselves might introduce errors, i.e. the system may not execute the given control parameters exactly due to inherent systematic imperfection such

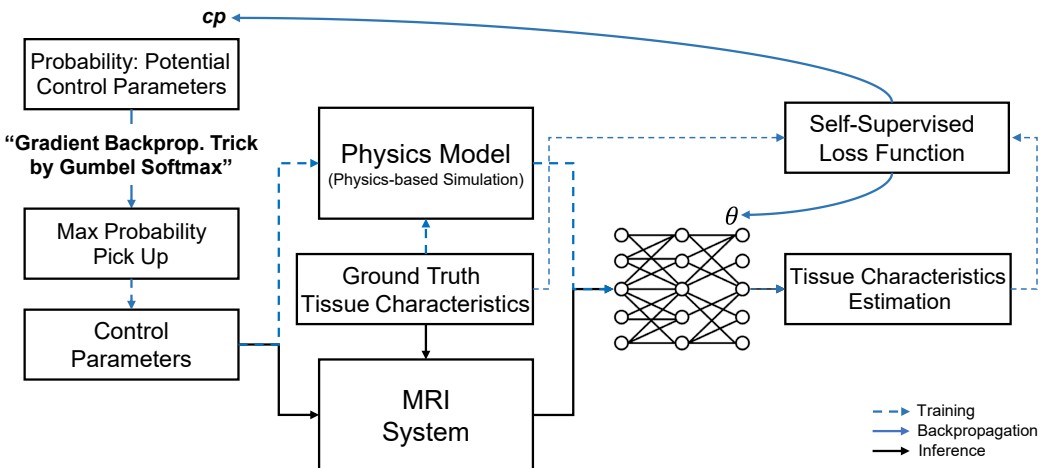

Figure 2: A schematic of the proposed Gumbel-Softmax-based LOAS (GLOAS) framework for control parameters. Multiple sets of control parameters manipulate the system to generate signal modulations, which are used by a neural network to estimate tissue characteristics. Since the MRI system can be modeled with a physics-based model, signal modulations can also be generated through physics-based simulations for tissue characteristic estimations. The estimation error is calculated by comparing the results to ground truth values. This error is backpropagated using the gradient method to simultaneously update both the control parameters and the neural network. Note that all potential control parameters are explored at each iteration, and only the parameters with the highest probability would be selected via Gumbel-Softmax trick.

as field inhomogeneity, eddy current, gradient nonlinearity, etc (Krupa & Bekiesińska-Figatowska (2015); Graves & Mitchell (2013)). These error and noise should be also considered in the model:

$$\widehat{tc} = PM^{-1}(\mathbf{S}(tc, \mathbf{cp} + \delta\mathbf{cp}) + \mathbf{N}(\sigma)),\ \delta\mathbf{cp} = \mathbf{cp} * \mathbf{N}(\sigma) \tag{5}$$

where $\delta\mathbf{cp}$ is the system error for control parameters and $\mathbf{N}$ is the noise originated from the system each of which is assumed to be the additive white Gaussian noise with the noise level of $\sigma$ in our study.

The ability to discriminate between signal modulations for different tissue characteristics is determined by the control parameters. In other words, the choice of control parameters ($\mathbf{cp}$) influences the difficulty of inverse problem for estimating tissue characteristics ($tc$) from observed signal modulations ($\mathbf{S}$). The difficulty of inverse problem can change with the number of acquisitions ($N$) for the same model. Smaller $N$ would complicate the inverse problem whereas the larger $N$ would make it easier to solve the inverse problem.

## 2.2 LEARNING-BASED OPTIMIZATION OF CONTROL PARAMETERS

The overall algorithm of learning-based optimization of acquisition schedule (LOAS) is described in Algorithm 1. The MRI acquisition schedule consist of multiple sets of control parameters. A deep neural network is designed to solve the inverse problem of physics model for tissue characteristic estimation and thus the estimation errors with respect to the control parameters is calculated as follows (Kang et al. (2022); Perlman et al. (2022); Cohen & Otazo (2023)):

$$Loss(\theta, \mathbf{cp}) = \left|\left|\widehat{tc} - tc\right|\right|_2^2 = ||f_\theta(\mathbf{S}(tc, \mathbf{cp})) - tc||_2^2 \tag{6}$$

The backpropagation with gradient descent allows the control parameters to be updated to minimize estimation errors of tissue characteristics.

$$\mathbf{cp}_{i+1} = \mathbf{cp}_i - \gamma_{cp} \frac{\partial Loss(\theta, \mathbf{cp})}{\partial \mathbf{cp}} \tag{7}$$

$$\theta_{i+1} = \theta_i - \gamma_\theta \frac{\partial Loss(\theta, \mathbf{cp})}{\partial \theta} \tag{8}$$

where $\mathbf{cp}_i$ is the updated control parameters at $i^{th}$ iteration and $\gamma$ represents the learning rate. For each iteration, the control parameters ($\mathbf{cp}$) and the inverse-problem-solving neural network ($f_\theta$) are simultaneously updated. One million sets of tissue characteristics, accounting for possible scenarios, were utilized for optimization. For test dataset, ten thousand sets of tissue characteristics with ten times finer step size was used.

---

**Algorithm 1:** Learning-based Optimization of Control Parameters

    1. Randomly sample a batch of tissue characteristics ($tc$)

    2. Simulate signal modulations with the sampled tissue characteristics ($tc$) and randomly initialized control parameters ($\mathbf{cp}$) via physics model

    3. Generated signal modulations are fed to neural network ($f_\theta$) to solve the inverse problem for estimation of tissue characteristics.

    4. Estimated tissue characteristics ($\widehat{tc}$) are compared with the ground truth ($tc$) to calculate the estimation error ($Loss$).

    5. The error is backpropagated using gradient descent in order to update control parameter ($\mathbf{cp}$) and neural network ($f_\theta$), simultaneously.

    6. Iterate the step 1 to 5 until the error converges

---

## 3 PROBABILISTIC REPRESENTATION OF CONTROL PARAMETERS FOR OPTIMIZATION

### 3.1 CATEGORICAL REPARAMETERIZATION WITH GUMBEL-SOFTMAX

We use a categorical representation ($z$) for each control parameter ($cp$) to constrain the discrete min-max range, based on the prior knowledge of the hardware, along with their associated probabilities (Figure 3). Each bin corresponds to a respective candidate of control parameter and the number of bin determines the step size of control parameters. The Softmax function converts the representation into probability:

$$\pi_i = \frac{exp(z_i)}{\Sigma_{j=1}^{k} exp(z_j)}, \ i = 1, ..., k \tag{9}$$

where $\pi_i$ is a class probability for $i^{th}$ bin of discrete values. To obtain the value of max probability, a Gumbel-Max trick is often used to efficiently sample $y$ from a categorical distribution with class probabilities $\pi_l$ (Gumbel (1954); Maddison et al. (2014)).

$$y = \underset{l}{argmax} \left[ log\pi_l + g_l \right] \tag{10}$$

where $g_1, ..., g_k$ are i.i.d samples drawn from Gumbel (0, 1). However, sampling from $\pi$ with the Gumbel-max trick cannot compute the gradient due to the non-differentiability of argmax function. We adopt a Softmax function as a continuous differentiable approximation to enable the gradient calculation of sampling from the probabilities ($\pi$). Thus, the Gumbel-Softmax trick provides soft-labeled control parameters and then the control parameter with the maximum probability is selected as hard-labeled control parameters (Jang et al. (2017); Maddison et al. (2017)):

$$cp_{soft} = \frac{exp((log\pi_l + g_l)/\tau)}{\Sigma_{n=1}^{k} exp((log\pi_n + g_n)/\tau)}, \ l = 1, ..., k \tag{11}$$

$$cp_{hard} = max(cp_{soft}) \tag{12}$$

where $cp_{soft}$ is the soft-labeled control parameters, $cp_{hard}$ is the hard-labeled control parameters, and $\tau$ is a temperature. This reparameterization enables control parameters to be represented by probability of all possible candidates for each control parameter and their combinations. Additionally, $cp_{soft}$ can be smoothly annealed into a categorical distribution as the temperature decreases, enabling equal-probability exploration of all control parameter candidates with the high temperature at early iteration. The hard-labeled control parameters are used for physic-based simulation to generate the signal modulations whereas the soft-labeled control parameters are adopted for backpropagation (Figure 3).

In addition, discrete values for control parameters are more suitable for MRI system where hardware input values are usually discretized. For example, the step size of radio frequency (RF) saturation time ($T_{sat}$) is often 50ms due to its pre-defined block size of RF pulse and longer RF saturation is achieved by repetitively applying RF saturation blocks (Togao et al. (2016); Heo et al. (2019)). This requires rounding the conventionally optimized control parameters to the nearest available discrete values, which might not be optimal.

## 3.2 NOISE ROBUST GRADIENT DESCENT

To motivate the construction of the soft-labeled control parameters, we investigated the gradient from objective function ($Loss$) to soft-labeled control parameters ($\mathbf{cp_{soft}}$) under the chain rule:

$$\frac{\partial Loss(\theta, \mathbf{cp})}{\partial \mathbf{cp_{soft}}} = \frac{\partial Loss(\theta, \mathbf{cp})}{\partial \mathbf{S}} \frac{\partial \mathbf{S}}{\partial \mathbf{cp}} \frac{\partial \mathbf{cp}}{\partial \mathbf{cp_{soft}}} \tag{13}$$

The calculation of gradient is inaccurate in noisy real-world environments which may cause noise for stochastic gradient (Gitman et al. (2019)):

$$g = \nabla F + \eta, \quad where \, \mathbb{E}(\eta) = 0 \tag{14}$$

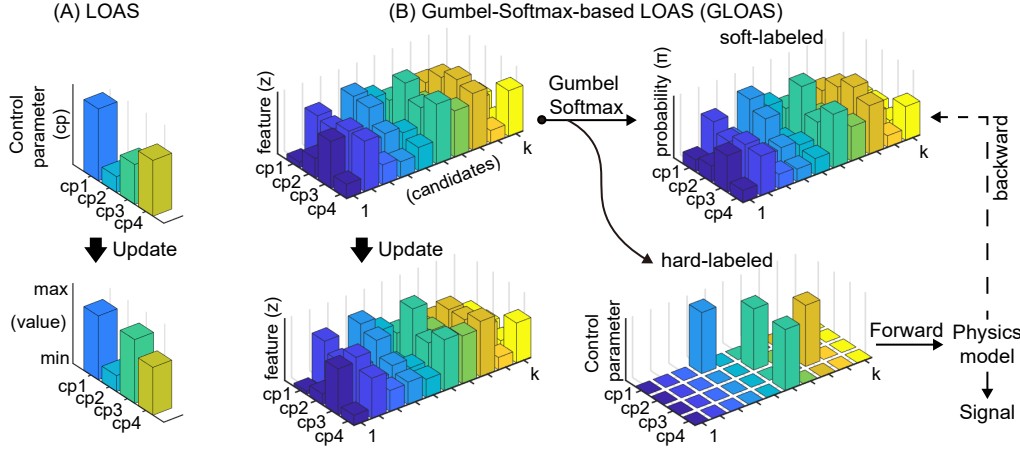

Figure 3: An illustration of control parameter updates for (A) the conventional LOAS and (B) the proposed Gumbel-Softmax-based LOAS (GLOAS) optimizations. For conventional LOAS, the values of control parameters are updated. For the proposed GLOAS, each control parameter is represented by multiple candidates (k) with probabilities to pick up the candidate of the highest probability via Gumbel-Softmax function. Hard-labeled control parameters are used for physics-based simulations, while soft-labeled control parameters are applied during backpropagation. Note that a single example set of control parameters is shown for illustration, but multiple sets ($N$) are required to generate signal modulations.

where $g$ is a stochastic gradient, $F$ is a objective function, and $\eta$ is a random noise of stochastic gradient. Therefore, the noise in signal modulations and control parameter can introduce a random noise for the gradient from $\mathbf{S} \in \mathbb{R}^{N \times 1}$ to $\mathbf{cp} \in \mathbb{R}^{NM \times 1}$:

$$\frac{\partial \tilde{\mathbf{S}}}{\partial \mathbf{cp}} = \frac{\partial \mathbf{S}}{\partial \mathbf{cp}} + \mathbf{E} \tag{15}$$

where $\tilde{\mathbf{S}}$ is a noisy signal modulation and $\mathbf{E}$ is a random interference matrix of gradient due to noise. The gradient is inaccurate for parameter updates in the conventional method, whereas the proposed soft-labeled control parameters ($\mathbf{cp_{soft}} \in \mathbb{R}^{NMc \times 1}$) modified the gradient as follows:

$$\frac{\partial \tilde{\mathbf{S}}}{\partial \mathbf{cp_{soft}}} = \frac{\partial \tilde{\mathbf{S}}}{\partial \mathbf{cp}} \frac{\partial \mathbf{cp}}{\partial \mathbf{cp_{soft}}} = (\frac{\partial \mathbf{S}}{\partial \mathbf{cp}} + \mathbf{E})\frac{\partial \mathbf{cp}}{\partial \mathbf{cp_{soft}}} = \frac{\partial \mathbf{S}}{\partial \mathbf{cp_{soft}}} + \Sigma_{i=1}^{NM} E_i G_i \tag{16}$$

where $E_i$ is a $i^{th}$ column vector of random interference matrix, $G_i$ is a $i^{th}$ row vector of Jacobian of Gumbel-Softmax function, $N$ is the number of acquisitions, $M$ is the number of control parameters for each acquisition, and c is the number of candidates for each control parameter. The proposed method not only calculates the gradients from loss function for possible candidates of control parameters, but also delivers more accurate gradients compared to the conventional approach in the presence of noise. The second term of the rightmost side of equation 16 would sum up to zero if $N \times M$ is high enough due to the randomness of the interference matrix. On the other hand, the gradient for the conventional optimization technique (equation 15) cannot reduce the random interference term making it susceptible to noise.

# 4 EXPERIMENTAL RESULTS

## 4.1 TISSUE CHARACTERISTIC ESTIMATION WITH NEURAL NETWORK

The goal of optimization of control parameters is to minimize the estimation error of tissue characteristics. Therefore, the estimation errors from the proposed GLOAS was compared to those from the conventional LOAS. As a case study, we adopted a two-pool proton exchange model as a physics model (*PM*), which is described with the modified Bloch-McConnell equations, to simulate the MR signal (See Appendix A).

We used a multi-layer perceptron (MLP) with seven hidden layers of 256 units each with ReLU activation for estimation network. The normalized tissue characteristics were obtained with the sigmoid activation function at the final layer, which was then re-normalized to each range. We performed optimizations on a dataset comprising of one million sets of four tissue characteristics, each of which was uniformly sampled from a pre-defined range (Table 2). The adaptive moment estimation (ADAM) optimizer was used to update the network and control parameters via backpropagation. Learning rates of estimation network ($\theta$) and control parameters ($\mathbf{cp}$) were heuristically determined as follows: $10^{-4}$ and $10^{-4}$ for the LOAS method and $10^{-4}$ and $10^{-2}$ for the GLOAS method. The temperature for the soft-labeled control parameters is annealed using the schedule $\tau = \max (1.0, 10\exp(-10^{-3t}))$ of the iteration step $t$.

The estimation errors of tissue characteristics were evaluated with respect to iteration for various conditions (Figure 4). The noise level and the number of acquisitions were changed. Basically, the difficulty of estimation for tissue characteristics increases as the number of acquisition decreases and the level of noise increases. Therefore, the most challenging condition is at the highest noise level ($\sigma = 0.02$) and the fewest acquisitions ($N$=10), whereas the easiest condition is without noise ($\sigma = 0$) and the largest acquisitions ($N$=30). This is well demonstrated in the Figure 4. The proposed GLOAS is on par with the LOAS optimization if the signal is free from noise. However, the GLOAS method outperforms the LOAS when noise are included in the physics-based simulation. The accuracy gap between the LOAS and GLOAS methods becomes larger as the complexity of inverse problem increases, i.e. reduction of the number of acquisitions and increase of the noise level. This results underscore the capability of the soft-labeled control parameters in calculating accurate gradients in the presence of noise (equation 16). In addition, the LOAS optimization resulted in high variance of estimation loss showing the instability of optimization and the dependency on

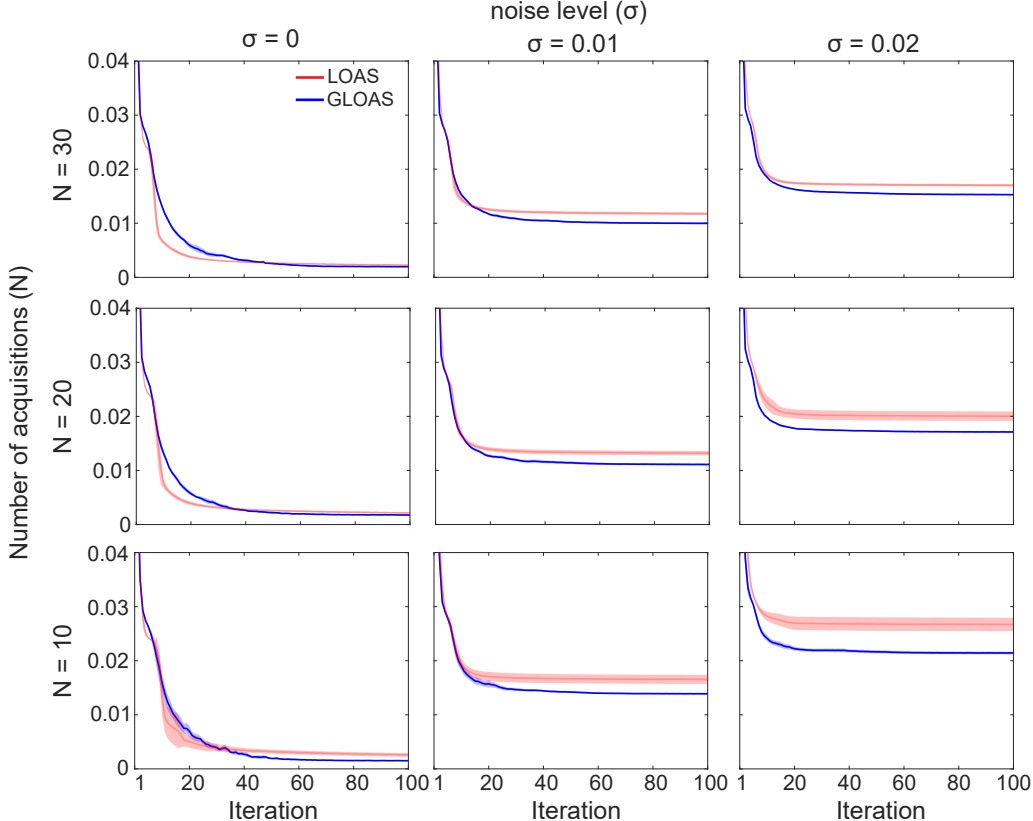

Figure 4: Training loss for tissue characteristic estimation as a function of iteration number, comparing the conventional LOAS and the proposed GLOAS approaches. Both optimizations were performed ten times to assess variance, representing the 95% confidence interval with the shaded regions. Results are shown for various levels of Gaussian noise and number of acquisitions which determine the difficulty of the inverse problem for tissue characteristic estimations.

initialization. The variation of estimation loss seems to be also dependent on the complexity of the inverse problem, showing the highest variance at the highest noise level and the fewest acquisitions. On the other hand, the GLOAS optimization provided relatively small variance of estimation errors, ensuring the stability of optimization process.

Table 1: Quantitative evaluation of tissue characteristic estimation using the LOAS and GLOAS methods at a noise level ($\sigma$) of 0.01 with various numbers of acquisitions. The normalized mean square errors (nRMSE) are reported for each tissue characteristic ($tc$). Each optimization was performed ten times to assess the variance. The mean and standard deviation were averaged over the last ten iterations.

| nRMSE | $N = 10$ | | 20 | | 30 | |
|---|---|---|---|---|---|---|
| (%) | LOAS | GLOAS | LOAS | GLOAS | LOAS | GLOAS |
| $tc1$ | $22.13 \pm 0.80$ | $20.13 \pm 0.20$ | $20.15 \pm 0.52$ | $18.57 \pm 0.37$ | $19.18 \pm 0.43$ | $17.82 \pm 0.40$ |
| $tc2$ | $11.14 \pm 0.56$ | $10.62 \pm 0.15$ | $9.88 \pm 0.21$ | $9.39 \pm 0.18$ | $9.36 \pm 0.21$ | $8.83 \pm 0.15$ |
| $tc3$ | $6.72 \pm 0.29$ | $6.17 \pm 0.32$ | $5.49 \pm 0.41$ | $4.59 \pm 0.21$ | $4.91 \pm 0.40$ | $3.96 \pm 0.17$ |
| $tc4$ | $2.67 \pm 0.36$ | $2.80 \pm 0.26$ | $1.96 \pm 0.16$ | $2.22 \pm 0.19$ | $1.81 \pm 0.22$ | $1.95 \pm 0.20$ |
| $mean$ | $10.67 \pm 0.50$ | $\mathbf{9.93 \pm 0.23}$ | $9.37 \pm 0.33$ | $\mathbf{8.69 \pm 0.24}$ | $8.82 \pm 0.31$ | $\mathbf{8.14 \pm 0.23}$ |

The test results for each tissue characteristic are also shown in Table 1. Overall, the normalized mean square error (nRMSE) values are lower with the GLOAS than those with the LOAS. However,

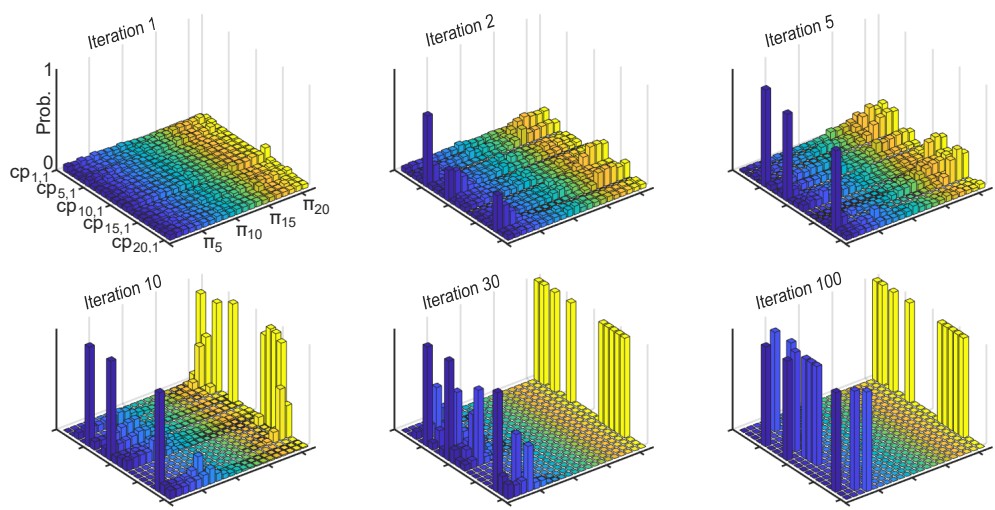

Figure 5: The probabilities for each candidate of the control parameter are displayed across different iterations. The sum of probabilities of all candidates for $cp_{i,1}$ is 1 and $cp_{i,1}$ is independent of each other. The acquisition number ($N$) of 20 and the noise level ($\sigma$) of 0.01 were used for optimization.

for $tc4$, the proposed method does not offer a big gain. This is presumably due to the inherently low difficulty of estimating $tc4$. Since the estimation error of $tc4$ is already low, the objective function, which sums the errors across all tissue characteristics, might focus on those with higher errors. In addition, the estimation of $tc1$ is challenging due to the low sensitivity of signal modulation with respect to change in $tc1$, which increases its vulnerability to noise (Kang et al. (2021)).

The simulation results demonstrate that the accuracy of tissue characteristic estimation is higher with the GLOAS method than those with the LOAS method under noisy condition. Therefore, the optimized control parameters from GLOAS has a potential to accelerate the temporal process by reducing the number of acquisitions without compromising the estimation accuracy. Especially, the acceleration of data acquisition is very important in the field of medical imaging to improve the patient comfort and reduce the artifacts caused by patient motions.

### 4.2 PROBABILISTIC REPRESENTATION OF CONTROL PARAMETERS

We assessed the probability of the soft-labeled control parameters during the optimization process. The probabilities of all candidates for each control parameter are monitored. As shown in Figure 5, at early iteration, the probabilities of all candidates are similar indicating that all possible control parameters are being equally considered. As iteration progresses, the probability of one candidate converges to one, while the probabilities of other candidates approach zero. This trend holds for every control parameter although the rates of convergence are may vary slightly. It is worth noting that optimized values often appeared at either the lower bound or the upper bound of the predefined control parameter ranges, which is consistent with previous findings (Zhao et al. (2018); Kang et al. (2022)). This is a commonly observed behavior in bang-bang control problems and mitigating this may be an important direction for future studies (Seyde et al. (2021)).

### 4.3 *In silico* TISSUE CHARACTERISTIC MAPS

The performance of the GLOAS optimization method was evaluated using the modified Brainweb-based digital phantoms simulated with two-pool proton exchange models. Four tissue characteristic maps were generated with previously reported gray matter (GM) and white matter (WM) values (Kim et al. (2020)). The four tissue characteristic maps were used to generate synthetic MRI images via physics model (eq24) with six schedules of acquisition number 10: three schedules were optimized with the GLOAS method and other three schedules were optimized with the LOAS method,

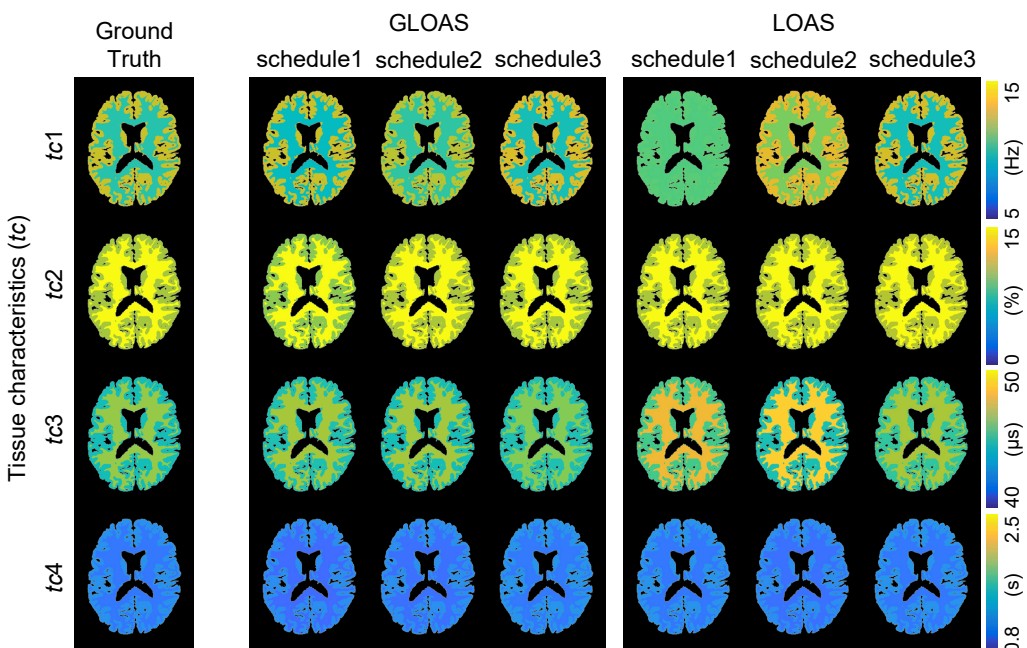

Figure 6: Physics-model-based custom-modified Brainweb digital phantom studies with multiple optimized schedules from the GLOAS and LOAS methods. Each schedule produces own synthetic MRI images via physics model and the synthesized MRI images were fed to the corresponding inverse-problem-solving neural networks simultaneously trained through optimization to estimate the tissue characteristic (*tc*) maps.

respectively. The synthetic MRI images were fed to the inverse-problem-solving neural network, simultaneously trained in optimization process, to estimate the tissue characteristic maps. As shown in Figure 6, the estimated tissue characteristic maps with optimized schedules from the GLOAS provide consistent results showing good agreements with the ground truth maps, whereas the LOAS showed unreliable results especially for $tc1$, which has an intrinsically low sensitivity. Given that the no noise was added to the synthetic MRI images, the GLOAS could allow an overall improved search for finding optimal control parameters by solving complex inverse problem of tissue characteristics estimation.

## 5 DISCUSSION

The main contribution of this work is the categorical reparameterization of control parameters with the Gumbel-Softmax trick which allows a probabilistic search over complex multidimensional space of physics model. The Gumbel-Softmax function enabled a selection of control parameters with the highest probability from the probabilistic representation so that the probability of control parameters is updated for optimization. Unlike the conventional LOAS optimization which uses a single set of control parameters for update, the proposed GLOAS calculates the gradients of numerous combinations of control parameters simultaneously. We demonstrated that the probabilistic representation was effective on optimization of control parameter, outperforming the LOAS framework in terms of the estimation accuracy of tissue characteristics and repeatability of optimization. In specific, we validated that the proposed GLOAS provides accurate stochastic gradient in the presence of noise. In addition, we showed that the proposed probabilistic optimization explores the numerous combinations of control parameters equally from early iteration which allows a comprehensive search over complex multidimensional space. Therefore, the proposed Gumbel-Softmax-based optimization could be an efficient tool for optimizing control parameters in various physical systems.

ACKNOWLEDGMENTS

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

## A BLOCH EQUATIONS FOR TWO-POOL PROTON EXCHANGE MODEL

The behavior of spins in MRI system can be described with Bloch equations. In specific, a two-pool proton exchange model, consisting of free bulk water pool (w) and the semisolid macromolecule pool (m), can be described with the modified Bloch-McConnell equations in the presence of radio frequency (RF) saturation. The signal originates from the protons in water so that the magnitude of net spin in longitudinal direction, which is equivalent to the longitudinal magnetization ($M_z$), in water pool is of interest (Henkelman et al. (1993); Heo et al. (2016)):

$$M_z^w(t) = \left(M_0^w - M_{ss}^w\right) e^{\lambda t} + M_{ss}^w \tag{17}$$

$$M_{ss}^w = M_0^w \frac{\frac{1}{T_1^m}\left(k_{mw} M_0^m T_1^w\right) + \alpha}{\left(k_{mw} M_0^m T_1^w\right)\left(Absor_{RF,m} + \frac{1}{T_1^m}\right) + \alpha\left[1 + \left(\frac{\omega_1}{2\pi\Omega}\right)^2\left(\frac{T_1^w}{T_2^w}\right)\right]} \tag{18}$$

$$\lambda = -\frac{1}{2}\left(\alpha + \beta - \sqrt{(\alpha - \beta)^2 + 4k_{mw}^2 M_0^m}\right) \tag{19}$$

$$\alpha = \frac{1}{T_1^m} + k_{mw} + Absor_{RF,m} \tag{20}$$

$$\beta = \frac{1}{T_1^w} + k_{wm} + Absor_{RF,w} \tag{21}$$

$$Absor_{RF,i} = \frac{\omega_1^2 T_2^i}{1 + (2\pi\Omega T_2^i)^2} \tag{22}$$

where $M_z^i$ is the longitudinal magnetization of a pool i; $M_0^i$ is the equilibrium magnetization of a pool i; $M_{ss}^w$ is the steady-state longitudinal magnetization of a pool i; $T_1^i$ and $T_2^i$ are the longitudinal and transverse relaxation times of a pool i, respectively; $\Omega$ is the frequency offset of the RF saturation; $\omega_1$ is the RF saturation amplitude; $k_{ij}$ is the proton exchange rate from a pool i to a pool j; and $Absor_{RF,i}$ is the RF absorption rate of a pool i. According to eq18, the signal originated from water pool is determined by three control parameters ($cp$): RF saturation power ($B_1 = \omega_1/2\pi\gamma$; $\gamma$ is the gyromagnetic ratio), frequency offset ($\Omega$), and saturation time (t = $T_{sat}$). A relaxation delay time (Td) is additionally defined to consider the recovery of the longitudinal magnetization in the absence of RF saturation which determines the initial longitudinal magnetization for next acquisition. Therefore, the final signal ($S$) can be described as follow:

$$S = \left[M_0^w\left(1 - e^{-Td/T_1^w}\right) - M_{ss}^w\right]e^{\lambda T_{sat}} + M_{ss}^w \tag{23}$$

$$tc = [k_{mw}, M_0^m, T_2^m, T_1^w], \quad cp = [B_1, \Omega, Ts, Td] \tag{24}$$

The ranges of tissue characteristics ($tc$) and control parameters ($cp$) are shown in the Table 2. According to the step size of control parameters, the numbers of candidates are 21 for $cp1$, 43 for $cp2$, 33 for $cp3$, and 21 for $cp4$. The lower and upper bounds of control parameters were constrained by the hardware configurations, clinical limitations, and properties of tissues. For example, the limited range of the RF saturation power was used to stay within the clinically permitted specific absorption rate (SAR), mainly due to the use of the SAR-intensive, time-interleaved parallel transmission (pTX)-based RF saturation (Togao et al. (2016); Heo et al. (2019)).

Table 2: Properties of control parameters ($cp$) and tissue characteristics ($tc$)

| | $cp1$ $B_1$ ($\mu$T) | $cp2$ $\Omega$ (ppm) | $cp3$ $T_{sat}$ (s) | $cp4$ $T_d$ (s) | $tc1$ $k_{mw}$ (Hz) | $tc2$ $M_0^m$ (%) | $tc3$ $T_2^m$ ($\mu$s) | $tc4$ $T_1^w$ (s) |
|---|---|---|---|---|---|---|---|---|
| max | 1.9 | 50 | 2.0 | 4.5 | 100 | 17 | 100 | 3.0 |
| min | 0.9 | 8 | 0.4 | 3.5 | 5 | 2 | 1 | 2.0 |
| step size | 0.05 | 1 | 0.05 | 0.05 | 1.0 | 0.1 | 1.0 | 0.03 |

