# OpenReview forum: "A Probabilistic Approach to Optimizing Hardware Control Parameters in System Property Estimation using Gumbel-Softmax"
_ICLR.cc/2025/Conference — ICLR 2025 Conference Withdrawn Submission_

### Official Review · Reviewer_vJMp · 2024-10-25

**Soundness:** 2
**Presentation:** 2
**Contribution:** 2
**Rating:** 3
**Confidence:** 4

**Summary:**

This paper applies the Gumbel-softmax reparameterization trick to optimize the hardware control parameters. It takes the MRI system as an example, demonstrating that the proposed method can achieve lower estimation errors than the conventional approach.

**Strengths:**

This paper is well-structured and considers a realistic problem. The application of the Gumbel-softmax reparameterization trick sounds correct in the context.

**Weaknesses:**

- The novelty of this paper is limited, as the Gumbel-Softmax trick has been implemented to address the categorical distribution problem in many fields over the past few years. Besides merely applying the trick to another concrete problem, additional improvements and contributions are expected.
- In the evaluation, comparing with only one other method is not comprehensive. More SOTA methods of optimizing the control parameters should be mentioned in the introduction and compared with the proposed one.
- The notations in the equations are inappropriate, such as in (4), (5), etc. Using letters (such as Greek letters) instead of abbreviations would be more suitable.
- Some training details are ignored such as the initialization of the unnormalized log-probabilities (logits), and whether the Gumbel noise was added once or at each iteration.

**Questions:**

- Is the temperature annealing necessary? And why is the starting temperature set to be $1$ but not greater than $1$?
- Figure 3 shows the training loss decreasing from $4\times 10^{-7}$ to around $1\times 10^{-7}$, why is the reduction so minimal? The errors summarized in Table 1 approximate $10$, why are these values so greater than the losses?
- Figure 4 shows the selected parameters at the end are all at both extremes, why were none of the parameters in the middle selected? Are the results totally different from those from the conventional method? After training, did the other logits $z$ become zeros or negative values?

---

> ### Author Response · Authors · 2024-11-22
>
> We would like to thank the reviewer for thoughtful and constructive comments. Our responses for each comment are written.
>
> Weakness.
>
> Although the Gumbel-Softmax trick has been implemented to address the categorical distribution problem in many fields, it has not been explored for control parameter optimization in MRI field. One of our main argument is that leveraging categorical distributions of MRI control parameters (acquisition parameters) is crucial in their optimization process, because MRI system works with discrete acquisition parameters, e.g. the step size of RF saturation time is often 50ms due to its pre-defined block size of RF pulse and longer RF saturation is achieved by irradiating RF saturation blocks repetitively. To the best of our knowledge, this is the first work that applies categorical reparameterization of control parameters for MR acquisition optimization and we demonstrated that our approach achieves noise-robustness and independency on randomly initialized control parameters during optimization process. Learning-based optimization of acquisition schedules (LOAS) has been shown to outperform other optimization methods based on indirect objective functions in previous work [1]. Our study aims to enhance the LOAS framework by incorporating categorical distributions. Accordingly, we focus on comparing the proposed method with the conventional LOAS approach. This clarification has been included in the introduction.
> Moreover, to clarify the novelty of the proposed method, we changed the title of the paper to “A probabilistic approach to optimizing acquisition scheduling of MRI using Gumbel-softmax”.
>
> Q1.
> We employed a high temperature during the early iterations to enable equal-probability exploration of all control parameter candidates, gradually decreasing the temperature to smoothly anneal into a categorical distribution. When the temperature is fixed to 1, the probability of control parameters converges to a categorical distribution quickly which implies that optimization becomes highly dependent on the specific candidate. We explained this in the Section 3.1.
>
> Q2.
> Thank you for pointing out the wrong magnitude of the training loss. We removed the x10-5 from the Figure 3. Since the training loss is MSE and the nRMSE values are shown in Table1, the magnitude value should be similar (only root difference).
>
> Q3.
> This is a very important question. The optimal values often appeared at either the lower bound or the upper bounds of the predefined control parameter ranges, which is consistent with the findings from previous works [1,2]. This behavior is commonly observed in Bang-Bang control problems. Mitigating this may be an important direction for future studies. We clarified this in the Section 4.2.
> After training, the logits 𝑧 corresponding to non-optimal discrete values become negative, as there is no constraint requiring the logits 𝑧 to remain positive.
>
> Minor Question in Weakness.
> The initial unnormalized log-probabilities (logits) were randomly determined and the Gumbel noise was added at each iteration.
>
>
> 1. Kang, B., Kim, B., Park, H., & Heo, H. Y. (2022). Learning‐based optimization of acquisition schedule for magnetization transfer contrast MR fingerprinting. NMR in Biomedicine, 35(5), e4662.
> 2. Zhao, B., Haldar, J. P., Liao, C., Ma, D., Jiang, Y., Griswold, M. A., ... & Wald, L. L. (2018). Optimal experiment design for magnetic resonance fingerprinting: Cramér-Rao bound meets spin dynamics. IEEE transactions on medical imaging, 38(3), 844-861.

---

> > ### Comment · Reviewer_vJMp · 2024-11-25
> >
> > Thank you for responding to the questions, which should be a good start to polish the paper. However, I still believe that the submitted version of the paper requires significant revisions/improvements in terms of presentation and evaluation. More importantly, there is the issue of novelty. While I agree with the authors that the proposed method is novel for the specific problem addressed, its novelty in the context of **Learning**, particularly given the scope of ICLR, is quite limited. Therefore, I consider this paper to fall below the acceptance threshold for ICLR.

---

### Official Review · Reviewer_dmPg · 2024-11-02

**Soundness:** 3
**Presentation:** 2
**Contribution:** 2
**Rating:** 3
**Confidence:** 5

**Summary:**

In this work, the authors propose to optimize the hardware control parameters with deep learning approach. Generally saying, this is an interesting application of AI4Physics by fitting the physical procedure with neural network.

**Strengths:**

1. The topic is interesting.

2. The writing is easy to follow.

**Weaknesses:**

1. While the framework is quite general, the experiments are not sufficient. Is there any dataset from industry or from the simulation of more integrated system?

2. What does it mean for the differences shown in Figure 3? How to evaluate the practical differences caused by such fitness difference?

3. The description of the physical procedure could be shorten as well as the definition of non-specific formula like 9-11. The authors should focus on how and why the NN improves the overall estimation.

4. As there are many other approaches and models in fitting parameters like evolutionary algorithm and other data fitting approaches. If this problem is a worthy one, there should be existing results to allow for the comparison.

**Questions:**

Please check the weakness.

---

> ### Author Response · Authors · 2024-11-22
>
> We would like to thank the reviewer for thoughtful and constructive comments. Our responses for each comment are written.
>
> Q1.
> As the reviewer commented, the proposed framework seems to be too broad without the experiments supporting its generalizability to other systems. Therefore, we revised the title of the paper to narrow down the scope specifically to the quantitative MR imaging as follows: “A Probabilistic Approach to Optimizing MRI control parameters using Gumbel-Softmax”
> In quantitative MRI, especially in MR fingerprinting field, the acquisition schedule optimization plays a critical role, but no open source dataset is available. The experimentally acquired MR images using an optimal schedule are essential for validating the effectiveness of the optimization process. Nevertheless, the reliability of physics-based models in the MRI field—especially the two-pool proton exchange model used in our study—has been well-established in previous studies, which demonstrated excellent agreement between experimentally observed signals and simulated signals [1,2].  This ensures that the optimized acquisition schedule derived from simulation data can be effectively translated to the in vivo data.
>
> Q2. The small error indicates the high accuracy of estimation of tissue characteristics. Once the optimization process is over, the optimal acquisition schedule for MRI and the corresponding inverse-problem-solving neural network for tissue characteristic estimation are obtained. The real in vivo MRI data acquired with our optimal schedule would be fed to the neural network to estimate the in vivo tissue characteristic maps. To further show the effectiveness of the proposed method, we added the figure which illustrates the in silico quantitative tissue characteristic maps estimated from schedules optimized with the conventional LOAS and proposed methods, respectively (Figure 6).
>
> Q3.
> We explained the Gumbel-softmax reparameterization to be self-contained. Since the key idea of the proposed work is the categorical reparameterization of control parameters, we believe that brief overview of the Gumbel-softmax is essential for clear understanding.
>
> Q4.
> This is a very important question. As the reviewer commented, there are many methods that can estimate tissue characteristics. However, these fitting-based and dictionary-based approaches cannot be incorporated in the proposed framework due to their incapability to calculate the gradient for back-propagation. Moreover, it was shown that the optimal schedule from learning-based optimization scheme (LOAS) achieves excellent performance in estimation of tissue characteristics against various comparison studies [3]. Since we proposed an optimizing method for acquisition schedule that outperformed the previous LOAS method, the optimal schedule can be applied to any quantification method. We added more explanations about existing works in the introduction section.
>
> 1. Kim, B., Schär, M., Park, H., & Heo, H. Y. (2020). A deep learning approach for magnetization transfer contrast MR fingerprinting and chemical exchange saturation transfer imaging. Neuroimage, 221, 117165.
> 2. Heo, H. Y., Zhang, Y., Lee, D. H., Hong, X., & Zhou, J. (2016). Quantitative assessment of amide proton transfer (APT) and nuclear Overhauser enhancement (NOE) imaging with extrapolated semi‐solid magnetization transfer reference (EMR) signals: application to a rat glioma model at 4.7 Tesla. Magnetic resonance in medicine, 75(1), 137-149.
> 3. Kang, B., Kim, B., Park, H., & Heo, H. Y. (2022). Learning‐based optimization of acquisition schedule for magnetization transfer contrast MR fingerprinting. NMR in Biomedicine, 35(5), e4662.

---

> > ### Comment · Reviewer_dmPg · 2024-11-25
> > **Response to the rebuttal**
> >
> > As also pointed out by other reviewers, the analysis and significance of the current work still needs improvement. Specially for my question, if the physical simulation is reliable, why not boosting it directly instead of fitting with DL?

---

### Official Review · Reviewer_zeR2 · 2024-11-04

**Soundness:** 2
**Presentation:** 1
**Contribution:** 2
**Rating:** 3
**Confidence:** 2

**Summary:**

This paper proposes an optimization method for control parameter search in physics-based models. The method relies on the categorical reparametrization of the control parameters through Gumbel-Softmax, simultaneously enabling non-deterministic search across the control parameter space, and  improving the robustness of gradient descent under the presence of noise.
The method is evaluated on a two-pool proton exchange model using an MLP, evaluated under varying noise levels and dataset sizes. Compared to a "conventional" approach (that does not rely on the Gumbel-Softmax reparametrization), the proposed method demonstrates better robustness to noise and improved accuracy,

**Strengths:**

This paper presents a concise method for parameter search in noisy environments, demonstrating its potential on a complex physical system simulator. The simplicity of the proposed approach contributes to its robustness, making it promising for applications that demand accurate parameter estimation under noise and under a large search space.

Notably, the method simultaneously optimizes the parameters of a neural network, which estimates system properties, alongside the control parameters defining the physical system. This integrated approach appears well-suited to the challenges in the targeted application domain, addressing both noise and the inherent difficulty of optimizing control parameters in complex systems.

**Weaknesses:**

- The paper lacks clear contextualization relative to prior work, omitting key references to research on soft labels and reparameterization in search problems—well-studied subjects in ML and optimization.
- Validation is limited to a single physics-based model, narrowing its scope, with unclear implications for generalization to related systems. The exclusive use of simulated data in experiments further restricts claims of noise robustness.
- Additionally, the method is only compared to an undefined "conventional" approach, with no comparisons to other ML-based methods in the literature, making its contribution to the field unclear.

**Questions:**

- Can you clarify what are the characteristics of the  "conventional" method used as a baseline?
- Could you clarify how does this method extend to other use cases (physics-based models and beyond)?
- What do you see as the main contribution of the paper? Is it the overall optimization methodology or its applicability to the considered application?

---

> ### Author Response · Authors · 2024-11-22
>
> We would like to thank the reviewer for thoughtful and constructive comments. Our responses for each comment are written.
>
> Weakness.
>
> Our proposed work is based on the learning-based optimization of acquisition schedule (LOAS) framework that updates scan (control) parameters to directly minimize the tissue estimation error with the help of neural network [1]. The LOAS method outperformed other optimization methods which are based on indirect objective functions such as maximizing signal discrimination between tissue types and minimizing the variance of estimates using the Cramer-Rao bound [2,3]. Our work is to improve the LOAS framework by incorporating the categorical reparameterization. Therefore, we focus on the comparison between the proposed and conventional LOAS approaches. We clarified this in the introduction and changed “conventional” to “LOAS” in the main manuscript.
> We also acknowledge that the original title of the paper seems to be too broad without the experiments supporting its generalizability to other systems. Therefore, we changed the title of the paper to narrow down the scope to specifically to MRI as follows: “A Probabilistic Approach to Optimizing MRI control parameters using Gumbel-Softmax”
>
> Q1.
> As mentioned above, we clarified the conventional method as LOAS method.
>
> Q2.
> This is a very important question. If the physics-based model is available for the given physical system, the control parameters (acquisition parameters) can be optimized to effectively analyze the system. For MRI system, for instance, the acquisition parameters are optimized to produce accurate tissue characteristic maps. This approach is not limited to MRI and can be extended to other quantitative systems where observed signals depend on the system's control parameters, such as geothermal energy production systems, where reservoir properties are the primary focus [4].
>
> Q3.
> The main contribution of this paper lies in its applicability to physical systems where physics-based models can accurately simulate the system. Furthermore, to the best of our knowledge, this is the first work to apply categorical reparameterization of control parameters for MRI sequence optimization. To enhance the effectiveness of the proposed method, we have included an additional figure (Figure 6) showing the quantitative tissue parameter maps estimated using schedules optimized with both the conventional LOAS method and the proposed approach. As demonstrated by our experimental results, this work is capable of addressing real-world noisy systems effectively.
>
> 1. Kang, B., Kim, B., Park, H., & Heo, H. Y. (2022). Learning‐based optimization of acquisition schedule for magnetization transfer contrast MR fingerprinting. NMR in Biomedicine, 35(5), e4662.
> 2. Zhao, B., Haldar, J. P., Liao, C., Ma, D., Jiang, Y., Griswold, M. A., ... & Wald, L. L. (2018). Optimal experiment design for magnetic resonance fingerprinting: Cramér-Rao bound meets spin dynamics. IEEE transactions on medical imaging, 38(3), 844-861.
> 3. Cohen, O., & Rosen, M. S. (2017). Algorithm comparison for schedule optimization in MR fingerprinting. Magnetic resonance imaging, 41, 15-21.
> 4. Wang, N., Chang, H., Kong, X. Z., & Zhang, D. (2023). Deep learning based closed-loop well control optimization of geothermal reservoir with uncertain permeability. Renewable Energy, 211, 379-394.

---

> > ### Comment · Reviewer_zeR2 · 2024-11-25
> >
> > I thank the authors for their response. Addressing these and others reviewers' questions and comments will improve the quality and clarity of the paper. As mentioned in my review, the proposed method is highly relevant to the application scope considered in the paper. However, the significance and novelty of the main contribution is not broad enough for the target venue. Thus, my acceptance score remains unchanged.

---

### Official Review · Reviewer_P1bB · 2024-11-05

**Soundness:** 2
**Presentation:** 2
**Contribution:** 2
**Rating:** 3
**Confidence:** 3

**Summary:**

The manuscript discusses a methodology to optimally adjust the control parameters of a physical system specified by set of equations and  a set of system parameters. The idea is to simultaneously train a neural network system parameter estimator and optimize the control parameters of the system such that the estimated system parameters are as accurate as possible. Rather than maintaining a single estimated value per control parameter, the manuscript proposes to use a categorical distribution over a discretized range of the parameters.

A set of experiments are conducted on a system with 4 control parameters and 4 system parameters simulating magnetization based on modified Bloch equations.

**Strengths:**

* System inversion and parameter estimation is a relevant field of research.

**Weaknesses:**

* The experimental validation is not convincing. The authors use a single baseline "conventional optimization approach" to study their method. The simulated system is very basic.
* Some parts of the text could be improved: e.g. 1. Intro l. 40 "parameters is to find an", l. 42 "signal from physical system", l. 47 "Thus physics model enables". 2.1 Physics Model l.159 "the noise could be originated from"
* It is not simple to reproduce the experimental findings given the information provided in the manuscript.
* The manuscript needs to better connect to the big literature of system identification and parameter estimation.

**Questions:**

* Why do you have "hardware control parameters" in the title rather than "control parameters" given that in the experimental Section, in Equation (1) and Figure 1 you are studying a physics model without any hardware footprint? It took me a while to understand that the entire methodology is virtual, hence this might be misleading for other readers in particular since the term MRI features in the abstract.
* Do you agree that the paper doe no set any "optimal hardware control parameters for MRI system" as claimed in the Introduction and the phrase should be removed?
* Could you please move the relevant aspects of your spin system from the Appendix to the main text (Experiments) to facilitate reading?
* Do you intend to provide code for the methodology and the experiments?
* Is there an explanation why the training loss for the "conventional" method is smaller given that the Gumbel-parametrized model has less options due to the discretization of the parameter space?
* In a practical setting, how do you differentiate between system parameters and control parameters?
* Can you explain, why you cited [Black 1986] and [Blanter et al. 2000] in Section 2.1?
* Can you make your noise model more explicit by integrating it more prominently in Eq. (1)? In practice, it is impossible to observe a clean signal.
* Can you please explain the additive noise component in the control parameters, in particular in the Experiments section?
* How would you select a good discretization scheme in practice both in terms of lower/upper bound and step size?
* Is there a sweet spot the number of discrete steps? Possibly, very coarse discretizations allow little flexibility while overly fine discretizations might be hard to estimate.
* Can you characterize the systems where your idea is applicable and beneficial?
* What about strongly (anti)correlated sets of parameters? What about parameters with little influence on the system behavior?

---

> ### Author Response · Authors · 2024-11-22
>
> We would like to thank the reviewer for thoughtful and constructive comments. Our responses for each comment are written.
>
> Q1.
> As the reviewer commented, our optimizing model is conducted by physics-based simulation to find optimal MRI control parameters providing the lowest estimation error of tissue characteristics. We used the term ‘hardware control parameters’ to refer that it is acquisition parameters (e.g., repetition time) that changes MRI hardware system, however, we agree with reviewer’s comment that this term might be confusing. We changed the term “hardware control parameters” to “control parameters”. In addition, to clarify the novelty of the proposed method on MRI acquisition schedule optimization, we changed the title of the paper to “A Probabilistic Approach to Optimizing MRI control parameters using Gumbel-Softmax”.
> In the field of MRI, the validity of physics-based models has been well established in previous studies. In particular, the two-pool proton exchange model used in our study demonstrated excellent agreement between experimentally observed signals and simulated signals. [1,2]. This ensures that simulation-based studies can be well translated into real MRI systems.
> In addition, we included a new figure (Figure 1) to enhance the readability of the paper, particularly for readers unfamiliar with quantitative MRI, where optimizing control parameters is crucial for accurately estimating tissue characteristic maps.
>
> Q2.
> This can be addressed by narrowing down the scope of the paper for MRI system.
>
> Q3.
> In MRI, there are many contrasts and quantitative MRI methods for different purpose. Each method is based on a different physics model, and any model can be incorporated into our proposed framework for optimizing the control parameters. To prevent any misunderstanding that our method is limited to specific MRI applications, we included an explanation of the spin system in the Appendix.
>
> Q4.
> We will make the code available upon publication.
>
> Q5.
> Basically, solving the inverse problem with a finite number of MRI acquisitions (measurements) is challenging, and this becomes even more difficult in the presence of noise.
> As we explained in the main manuscript, the proposed Gumbel-Softmax-based optimization is on par with the conventional optimization if the signal is free from noise. However, the proposed method outperforms the conventional method when noise is included in the physics-based simulation. The accuracy gap between the conventional and the proposed methods becomes larger as the complexity of inverse problem increases. This is presumably due to the noise robust gradient descent of the proposed optimization as claimed in Section 3.2. Since the interference matrix is generated by random noise in the gradient, each interference matrix is random and independent of the others, whereas the Jacobian of the Gumbel-Softmax function (G) remains constant. Consequently, the term 𝐸𝑖𝐺𝑖 can be treated as a random matrix that averages out to zero according to the central limit theorem.
>
> Q6.
> In MRI system, system properties are the tissue characteristics that we aim to estimate, while the control parameters are the acquisition settings that can be adjusted in the MRI console. Particularly, we had four degree of freedom for acquisition settings in our study which are: saturation time of RF pulse, saturation power of RF pulse, off-resonance frequency offset of RF pulse, and relaxation delay time without RF pulse. As the reviewer commented, there might be additional system parameters that we cannot control over and this is not a subject of optimization, because they do not influence the estimation of tissue characteristics.

---

> ### Author Response · Authors · 2024-11-22
>
> Q7.
> We wanted to show that many systems result in different kind of noise distributions, however, as of now that we narrowed down our scope to MRI system, we removed them to avoid confusion.
>
> Q8.
> The Eq. (1) describes the physics model that can simulate the real physical system (MRI system). We can simulate the signal without noise, however, as the reviewer commented, noise is prevalent in any system and thus we explained this in Eq. (4).
>
> Q9.
> As explained in the Section 2.1, the control parameter themselves might introduce errors, i.e. the system may not execute the given control parameters exactly due to inherent systematic imperfection. This commonly appears in MRI system as artifacts from field inhomogeneity, eddy current, and many others [3]. We added detailed explanation about aforementioned artifacts in the Section 2.1. Additionally, the noise in control parameters were assumed to be the additive white Gaussian noise with the noise level of σ in our study: Δcp = cp*N(σ). We clarified this in Section 2.1.
>
> Q10&Q11.
> The lower and upper bounds were constrained by the hardware configurations or clinical limitations. For example, the limited range of the RF saturation power was used to stay within the clinically permitted specific absorption rate (SAR), mainly due to the use of the SAR-intensive, time-interleaved parallel transmission (pTX)-based RF saturation. Similarly, the step size of RF saturation time is often 50ms due to its pre-defined block size of RF pulse and longer RF saturation is achieved by irradiating RF saturation blocks repetitively [4,5]. We added this in the Section 3.1.
>
> Q12.
> This can be addressed by narrowing down the scope of the paper for MRI system.
>
> Q13.
> This is a very important question. In quantitative MRI, only the control parameters that are correlated with the tissue characteristics to be estimated are varied to generate the signal, while other parameters remain fixed. Therefore, the choice of control parameters to be adjusted is also important and this is often performed based on the analysis of the physic model which has been intensively studied. As of today, the various MRI protocols have been established for different qMRI techniques [6].
>
> 1. Kim, B., Schär, M., Park, H., & Heo, H. Y. (2020). A deep learning approach for magnetization transfer contrast MR fingerprinting and chemical exchange saturation transfer imaging. Neuroimage, 221, 117165.
> 2. Heo, H. Y., Zhang, Y., Lee, D. H., Hong, X., & Zhou, J. (2016). Quantitative assessment of amide proton transfer (APT) and nuclear Overhauser enhancement (NOE) imaging with extrapolated semi‐solid magnetization transfer reference (EMR) signals: application to a rat glioma model at 4.7 Tesla. Magnetic resonance in medicine, 75(1), 137-149.
> 3. Krupa, K., & Bekiesińska-Figatowska, M. (2015). Artifacts in magnetic resonance imaging. Polish journal of radiology, 80, 93.
> 4. Togao, O., Hiwatashi, A., Keupp, J., Yamashita, K., Kikuchi, K., Yoshiura, T., ... & Honda, H. (2016). Amide proton transfer imaging of diffuse gliomas: effect of saturation pulse length in parallel transmission-based technique. PloS one, 11(5), e0155925.
> 5. Heo, H. Y., Xu, X., Jiang, S., Zhao, Y., Keupp, J., Redmond, K. J., ... & Zhou, J. (2019). Prospective acceleration of parallel RF transmission‐based 3D chemical exchange saturation transfer imaging with compressed sensing. Magnetic resonance in medicine, 82(5), 1812-1821.
> 6. Cohen-Adad, J., Alonso-Ortiz, E., Abramovic, M., Arneitz, C., Atcheson, N., Barlow, L., ... & Xu, J. (2021). Generic acquisition protocol for quantitative MRI of the spinal cord. Nature protocols, 16(10), 4611-4632.

---

> > ### Comment · Reviewer_P1bB · 2024-11-26
> >
> > Many thanks for your response. Making the suggested changes will improve the manuscript. In the current state, my score remains unchanged.

---

### Official Review · Reviewer_wifG · 2024-11-07

**Soundness:** 2
**Presentation:** 2
**Contribution:** 2
**Rating:** 3
**Confidence:** 3

**Summary:**

This study aims to improve the accuracy of self-supervised estimation of system properties (sp) by estimating better the control parameters (cp) of hardware.
Proposed method is based on quantized representation of cp, and Gumbel-Softmax operation (Jang et al., 2017).
Authors claim that the proposed method enables a probabilistic search across an expanding set of all candidates for each control parameter, and it works better under noisy environments.

**Strengths:**

S1: The idea of quantizing cp may be interesting and promising for better search across an expanding set of all candidates.

S2: The paper is easy to read.

**Weaknesses:**

*W~ is critical.

W1: I'm curious to see if there are similar ideas of quantization (for better search across an expanding set of all candidates) in other, more general research areas. Namely, I cannot judge the degree of novelty of quantization from the description of the paper alone.

W2: Regarding the Gumbel-Softmax operation, the proposed method is a direct application of (Jang et al., 2017).
I think that a methodological novelty is not big.

*W3: I consider that the setting studied in this paper (sp is known, cp is unknown, and signal data is not available) does not simulate realistic situations.

W4: Explanation of the proposed method is not clear.

*W5: I cannot understand where the merits of the proposed method come from only with presented experimental results.
Also, experimental settings are not clearly explained or may not be appropriate.

W6: Minor optional points:

It may be confusing (abuse mathematically) that $^{−1}$ in $PM^{−1}(S(sp,cp))$ is only for sp.

$f$ is a deep neural network -> $f_\\theta$ is a deep neural network

$cp_{soft}$ in (11) -> $cp_{soft,l}$

`||` in (6) is strange -> use `\|` in Latex for $\\|\cdot\\|_2^2$

(7) and (8) require $|_{(\\theta,cp)= (\\theta_i,cp_i)}$ for partial derivatives.

I cannot understand ``One million sets of system properties, accounting for possible scenarios, were utilized for optimization. For test dataset, ten thousand sets of system properties with ten times finer step size was used.'' in Sec. 2.2.

equation 16 -> use `\eqref{}`

ADAM -> Adam (the latter is conventional)

normalized mean absolute error (nRMSE) -> normalized rooted mean squared error or nMAE

**Questions:**

*Q~ is critical.

Q1 (for W1):
It is my understanding that this paper claims that the proposed method's ability to successfully estimate cp has led to improved accuracy in estimating system properties.
If "the proposed method is useful for better search across an expanding set of all candidates" is true, wouldn't it be effective in a simpler regression setting?
Namely, I'm curious if there are similar discussions in a simpler regression setting?
Have authors researched that?
If such research exists, it could be used to make this paper more persuasive.
Also, is there a reason why quantization is effective in the setting of this paper specifically, and not in a regression setting?

Q2 (for W2):
Is W2 correct?

*Q3 (for W3):
Is W3 correct?
I consider this weakness to be the most serious.

*Q4 (for W3 and W5):
For example, when S=sp+cp, update (say, perturbation) of cp will just degrade self-supervised estimation accuracy of sp;
It will be better to keep cp fixed.
I think that the quantization of cp makes this situation more likely to occur.
Is there any point in estimating cp in the first place?

Q5 (for W4):
I cannot understand the necessity of Gumbel-Softmax operation around equations (10) and (11) only from this paper.
Is there any reason why we can't just use $\\pi_i$ in equation (9) for $cp_{soft,i}$ in (11)?
Is (10) efficient than "Sample $u\\sim Uniform(0,1)$, and let $y=\\min\\{l\\in\\{1,\\ldots,k\\}:\\sum_{i=1}^l\\pi_i>u\\}$"?
Or, is the temperature parameter $\\tau$ that is increasing in step $t$ important?

*Q6 (for W4):
In the proposed method, a quantized representation of the cp is used for learning, but it seems that a representative value of $cp$ is required to calculate $S(sp,cp)$.
The paper does not describe how to define that representative value.

Q7 (for W4):
Authors write "Categorical reparameterization of hardware control parameter is particularly beneficial for noisy real-world environments which may cause noise for stochastic gradient (Gitman et al. (2019)):"
Is this a claim of (Gitman et al. (2019)) or an authors' claim?
If this is an authors' claim, this writing is confusing, and I feel that an authors' paper does not properly provide a rational for this claim:
I cannot understand "The second term of the rightmost side of equation 16 would sum up to zero if $N\\times M$ is high enough due to the randomness of the interference matrix."

*Q8 (for W5):
In numerical experiments, I cannot understand which of the following was effective in causing the proposed method to outperform a conventional method: update of cp, quantization of cp, or Gumbel-Softmax operation.
Please conduct an ablation study: for example, compare the no-update method (fix cp at some values; see Q4), the conventional method (without quantization of cp), another method (with quantization of cp and without Gumbel-Softmax operation; see Q5), the proposed method (with quantization of cp and Gumbel-Softmax operation).

Q9 (for W5):
Is the computational efficiency of the proposed method comparable to other methods?
If the computational efficiency of the proposed method is poor, we can simply use another efficient method with a larger acquisition number $N$.

*Q10 (for W5):
I think the learning rate is quite important in this experimental comparison.
Did you try several different settings of the learning rate?
Did $(10^{-4},10^{-4})$ for the conventional method and $(10^{-4},10^{-2})$ for the proposed method yield the best result?
I am concerned about a situation where the iteration gets trapped in poor local minima since $(10^{-4},10^{-4})$ for the conventional method is too small.

Q11 (for W5):
I don't understand the significance of multiplying the sigmoid activation function in the final hidden layer.
If the scale issue is important, I think it would be better to standardize system properties themselves.

Q12 (for W5):
Why is the criterion (6) adopted for learning different from the criterion for evaluation (normalized mean absolute error or nRMSE in Table 1)?
If we want to improve the average normalized mean absolute error (or nRMSE), then our training criteria would be better to correspond to that.

---

> ### Author Response · Authors · 2024-11-22
>
> We would like to thank the reviewer for thoughtful and constructive comments. Our responses for each comment are written.
>
> Q1 (for W1). The model-based regression task fundamentally requires multiple measurements and focuses on identifying the optimal model that minimizes mapping errors. Independent measurements play a crucial role in achieving successful regression; hence, in real-world scenarios, multiple measurements are typically obtained by varying the system's control parameters. Here, our work mainly focuses on an optimization process to determine the best control parameters for accurate regression. This is the first work to expand the set of all potential candidates representing the system's control parameters. We applied our approach to determining optimal control (acquisition) parameters for an MRI system to quantify tissue characteristics through regression. In the method section, we provided mathematical analysis why our approach leads to better optimization. Additionally, we included a new figure (Figure 1) to enhance the readability of the paper, particularly for readers unfamiliar with quantitative MRI, where optimizing control parameters is crucial for accurately estimating tissue characteristic maps.
>
> Although the Gumbel-Softmax trick has been implemented to address the categorical distribution problems in many fields, it has not been explored for control parameter optimization in physics-based model. Especially, in MRI field, the learning-based optimization of acquisition schedule (LOAS) framework was recently proposed to provide optimal scheduling for control parameters [1,2], however, the LOAS method does not allow categorical search to update control parameters in estimating tissue characteristics (system properties). We clarified this in the introduction.
> Additionally, we changed the terminologies to enhance readability. Moreover, to clarify the novelty and focus of the proposed method, we revised the title of the paper to “A Probabilistic Approach to Optimizing MRI control parameters using Gumbel-Softmax”.
>
> One of our main argument is that leveraging categorical distributions of MRI control parameters is crucial in their optimization process, because MRI system works with discrete control parameters, e.g. the step size of RF saturation time is often 50ms due to its pre-defined block size of RF pulse and longer RF saturation is achieved by irradiating RF saturation blocks repetitively. The conventional LOAS method takes a continuous search and rounds the optimized control parameters for the nearest available discrete values, which might not be optimal. We clarified this in the Section 3.1.
>
> Q2 (for W2). As the reviewer commented, the proposed method is a direct application of categorical reparameterization for control parameters in MRI acquisition optimization. To the best of our knowledge, this is the first work that applies categorical reparameterization of control parameters for MR acquisition optimization and we demonstrated that our approach achieves noise-robustness and independency on randomly initialized control parameters during optimization process.
>
> *Q3 (for W3). The proposed method is based on the assumption that a physics-based model can accurately represent the real-world physical system. The validity of the two-pool proton exchange model has been established in previous studies, which demonstrated excellent agreement between experimentally observed signals and simulated signals [3,4].
>
> *Q4 (for W3 and W5). The whole framework of the proposed method is to obtain optimal cp (MRI acquisition parameters) that can maximize the estimation accuracy of sp (tissue characteristics). The model-based regression task inherently relies on multiple measurements and aims to identify the optimal model that minimizes mapping errors. Independent measurements are critical for successful regression; therefore, in practical applications, multiple measurements are often acquired by adjusting the system's control parameters (MRI acquisition parameters). So, fixed cp does not guarantee independent measurements and might result in inaccurate estimation of sp (tissue characteristics). Previous LOAS framework shows that updating optimal cp is achieved by self-supervised estimation accuracy.
>
> Q5 (for W4). Gumbel-softmax function enables the sampling of hard-labeled control parameters from soft-labeled control parameters with the inclusion of Gumbel sampling. We need to sampled one candidate with max probability in order to generate signal via physics model. The Gumbel-Softmax function allows the sampling of the maximum probability process to be backpropagated.
>
> Additionally, we employed a high temperature in the early iterations to ensure equal-probability exploration of all control parameter candidates and decrease the temperature to smoothly anneal into a categorical distribution. We add an explanation about the temperature in the Section 3.1.

---

> ### Author Response · Authors · 2024-11-22
>
> *Q6 (for W4).
> The representative value of control parameter is sampled through Gumbel-softmax reparameterization. Gumbel-softmax function enables the sampling of hard-labeled control parameters from soft-labeled control parameters with the inclusion of Gumbel sampling. The hard-labeled control parameters (cphard) can be calculated as follows: cphard = max(cpsoft). We added the equation in the main manuscript (Equation 12). This is implemented with the function torch.nn.functional.gumbel_softmax.
>
> Q7 (for W4).
> Thank you for pointing out the ambiguity. We revised the sentence to “The calculation of gradient is inaccurate in noisy real-world environments which may cause random noise error for stochastic gradient (Gitman et al. (2019)):”
> Since the interference matrix is generated by random noise in the gradient, each interference matrix is random and independent of the others, whereas the Jacobian of the Gumbel-Softmax function (G) remains constant. Consequently, the term 𝐸𝑖𝐺𝑖 can be treated as a random matrix that averages out to zero according to the central limit theorem.
>
> *Q8 (for W5).
> As explained in the response for Q4, the whole framework of the proposed method is to obtain optimal cp (MRI acquisition parameters) that can maximize the estimation accuracy of sp (tissue characteristics). The model-based regression task inherently relies on multiple measurements and aims to identify the optimal model that minimizes mapping errors. Fixed cp does not guarantee independent multiple measurements and results in inaccurate estimation of sp (tissue characteristics). In other words, if we fix the cp at specific values, only the neural network is updated to address the inverse problem of estimating sp (tissue characteristics) from insufficient measurements. As the estimation accuracy heavily depends on the configuration of the control parameters (CP), optimizing CP is crucial.
>
> Q9 (for W5).
> The computational efficiency for the conventional LOAS and proposed methods were similar. The optimization times were 6.5 hour for LOAS and 6.8 hour for proposed methods under our computational power (NVIDIA TITAN RTX GPU).
>
> *Q10 (for W5).
> We tested multiple settings of learning rate varying for (10-3, 10-4, 10-5, 10-6) for each optimizer and found that two methods showed best results with different combinations of learning rates.
>
> Q11 (for W5).
> Sigmoid activation enables stable learning where estimated sp (tissue characteristics; tc) have different scale. For example, T1 (tc4) has a range of seconds, but T2m (tc3) has a range of microseconds.
>
> Q12 (for W5).
> Any kind of loss (L1 or L2) can be used for the optimization. We adopted L2 in this study. The reason why we showed nRMSE values for each system property is due to the significantly different ranges of each system property (tissue parameters). We wanted to show normalized error for fair comparison among system properties.
>
> 1. Kang, B., Kim, B., Park, H., & Heo, H. Y. (2022). Learning‐based optimization of acquisition schedule for magnetization transfer contrast MR fingerprinting. NMR in Biomedicine, 35(5), e4662.
> 2. Perlman, O., Zhu, B., Zaiss, M., Rosen, M. S., & Farrar, C. T. (2022). An end‐to‐end AI‐based framework for automated discovery of rapid CEST/MT MRI acquisition protocols and molecular parameter quantification (AutoCEST). Magnetic Resonance in Medicine, 87(6), 2792-2810.
> 3. Kim, B., Schär, M., Park, H., & Heo, H. Y. (2020). A deep learning approach for magnetization transfer contrast MR fingerprinting and chemical exchange saturation transfer imaging. Neuroimage, 221, 117165.
> 4. Heo, H. Y., Zhang, Y., Lee, D. H., Hong, X., & Zhou, J. (2016). Quantitative assessment of amide proton transfer (APT) and nuclear Overhauser enhancement (NOE) imaging with extrapolated semi‐solid magnetization transfer reference (EMR) signals: application to a rat glioma model at 4.7 Tesla. Magnetic resonance in medicine, 75(1), 137-149.

---

> > ### Comment · Reviewer_wifG · 2024-11-23
> > **The 1st reply**
> >
> > W6:
> > It seems that very little was done about W6.
> > I feel that the presentation is still on the rating "2:fair" (not appealing).
> >
> > Q1 (for W1):
> > It seems that I did not convey my question correctly.
> >
> > This paper compares continuous representation (LOAS) and categorical representation (GLOAS) for cp-estimation in the self-supervised estimation problem (of sp).
> > I am interested in the comparison between continuous representation and categorical representation in a more simple "regression problem".
> > When estimationg $Y$ based on an observation of $X$ (regression problem), we can consider "direct estimation of $Y$" or "estimation of categorical representation of $Y$".
> > Have the authors investigated such studies?
> > I advised you that if such studies exist in previous studies, you can use them as material to support your research.
> >
> > Q2 (for W2): Okay.
> >
> > *Q3 (for W3): Okay.
> > This question is also related to Q4 and Q8.
> > See my replies regarding "*Q4 (for W3 and W5)", "Q5 (for W4): ", and *Q8 (for W5):".
> >
> > *Q4 (for W3 and W5):
> > See my reply regarding "Q5 (for W4): " and *Q8 (for W5):".
> >
> > Q5 (for W4):
> > I have understood Gumbel-Softmax trick, but I think my alternative procedure is possible.
> > The present writing style, which adopts the Gumbel-Softmax trick without any discussion as an example of various possible means to achieve your goal, does not serve the reader well.
> >
> > Also, the discussion of temperature parameters remains heuristic.
> > If optimization (or cp-estimation problem) were meaningful, the distribution $cp_{soft}$ would approach a spike distribution even if the temperature $\\tau$ was fixed.
> > I believe that by raising the temperature $\\tau$, the distribution $cp_{soft}$ was forced to approach a certain spike distribution, resulting in the situation described in Q4, which improved the estimation accuracy of your proposed method (because cp-estimation was meaningless).
> > See my reply regarding "*Q8 (for W5):", and take re-experiments.
> >
> > *Q6 (for W4):
> > It seems that I did not convey my question correctly.
> >
> > I understand that it involves a max operation like (12).
> >
> > $cp_{hard}$ is the label value corresponding to a certain interval $[a,b]$ of $cp$, but which value of $[a,b]$ is used as the representating value when calculating $S(sp,cp)$?
> > Is there no need for such a representating value?
> >
> > Q7 (for W4):
> > I understand that the claim "the calculation of gradient is inaccurate in noisy real-world environments which may cause noise for stochastic gradient" is of Gitman et al. (2019).
> >
> >
> > *Q8 (for W5):
> > I understand the authors' supporting hypothesis.
> > Report results of experiments to refute the negative hypothesis I have commented on.
> > I will not increase the rating at all unless you report a satisfactory re-experiment (please show me the experiment code if possible).
> >
> > Q9 (for W5): Okay.
> >
> > *Q10 (for W5):
> > You used $10^{-2}, 10^{-4}$ for learning rates for LOAS and GLOAS.
> > Why did you try $(10^{-3}, 10^{-4}, 10^{-5}, 10^{-6})$ in a re-experiment.
> > I think it is better to try $(10^{-1}, 10^{-2}, 10^{-3}, 10^{-4}, 10^{-5})$.
> >
> > Q11 (for W5): Okay.
> > I think that if the scale issue is important it would be popular to standardize system properties themselves.
> > But I do not persist in this issue.
> >
> > Q12 (for W5):
> > If you want to show nRMSE values for each system property for the evaluation due to the significantly different ranges of each system property (tissue parameters), you should use nRMSE loss for the optimization as well.

---

### Note · Authors · 2025-02-03

**Comment:**

I have read and agree with the venue's withdrawal policy on behalf of myself and my co-authors.

**Withdrawal Confirmation:**

I have read and agree with the venue's withdrawal policy on behalf of myself and my co-authors.